# The Alteration of CTNNBIP1 in Lung Cancer

**DOI:** 10.3390/ijms20225684

**Published:** 2019-11-13

**Authors:** Jia-Ming Chang, Alexander Charng-Dar Tsai, Way-Ren Huang, Ruo-Chia Tseng

**Affiliations:** 1Department of Surgery, Division of Thoracic Surgery, Chia-Yi Christian Hospital, Chiayi 60002, Taiwan; jameschang127@gmail.com; 2Department of Physical Therapy, College of Medical and Health Science, Asia University, Taichung 41354, Taiwan; 3Department of Molecular Biology and Human Genetics, Tzu Chi University, Hualien 97004, Taiwan; iwillneverknew@gmail.com; 4GLORIA Operation Center, National Tsing Hua University, Hsinchu 30013, Taiwan; g39604042@gmail.com

**Keywords:** CTNNBIP1, β-catenin, lung cancer

## Abstract

β-catenin is a major component of the Wnt/β-catenin signaling pathway, and is known to play a role in lung tumorigenesis. β-catenin-interacting protein 1 (CTNNBIP1) is a known repressor of β-catenin transactivation. However, little is known about the role of CTNNBIP1 in lung cancer. The aim of this study was to carry out a molecular analysis of CTNNBIP1 and its effect on β-catenin signaling, using samples from lung cancer patients and various lung cancer cell lines. Our results indicate a significant inverse correlation between the CTNNBIP1 mRNA expression levels and the CTNNBIP1 promoter hypermethylation, which suggests that the promoter hypermethylation is responsible for the low levels of CTNNBIP1 present in many lung cancer patient samples. The ectopic expression of CTNNBIP1 is able to reduce the β-catenin transactivation; this then brings about a decrease in the expression of β-catenin-targeted genes, such as matrix metalloproteinase 7 (MMP7). Conversely, CTNNBIP1 knockdown is able to increase β-catenin transactivation and the expression of MMP7. In agreement with these findings, a low level of CTNNBIP1 was found to be correlated with a high level of MMP7 when a publicly available microarray dataset for lung cancer was analyzed. Also, in agreement with the above, the ectopic expression of CTNNBIP1 inhibits the migration of lung cancer cells, whereas the CTNNBIP1 knockdown increases cancer cell migration. Our findings suggest that CTNNBIP1 is a suppressor of cancer migration, thus making it a potential prognostic predictor for lung cancer.

## 1. Introduction

Lung cancer is one of the leading causes of cancer deaths worldwide, including in Taiwan, with a five-year survival rate of less than 15%; this is despite significant advances in both the diagnostic and therapeutic approaches to this disease [1]. Therefore, it is a matter of urgency to explore in detail the precise biological properties of lung cancer, which will help with the development of better therapeutic modalities that are able to effectively treat lung cancer. Previously, our genome-wide loss of heterozygosity (LOH) study showed that a high frequency of LOH affecting the chromosomal region 1p36.2 was present in lung cancer patients [2]. Interestingly, this deleted region is close to the gene locus encoding the β-catenin-interacting protein 1 (CTNNBIP1) protein, which is a tumor suppressor-like protein that is known to be involved in regulating β-catenin signaling. However, its association with lung tumorigenesis has not yet been analyzed in detail.

β-catenin is a major component of the Wnt/β-catenin signaling pathway, and plays a dual role in vertebrates, in that it affects both the development and tumorigenesis [3]. In normal and non-stimulated cells, the majority of β-catenin protein is present at the cells’ adherens junctions with very little present in the cytoplasmic fraction or the nuclear fraction because of the protein’s rapid turnover [4]. The activated Wnt signaling causes β-catenin to accumulate in the cytoplasm, and the protein is then translocated to the nucleus, where it binds to T-cell factor/lymphoid enhancer-binding factor (TCF/LEF-1). This binding activates the expression of a range of target genes, including c-Myc and MMP7 [5,6,7]. Within the cell nucleus, the activity of the β-catenin/TCF complex is able to be inhibited by CTNNBIP1. This protein blocks β-catenin signaling by competing with TCF for binding to β-catenin. This in turn affects the transcription of the various target genes within the Wnt/β-catenin pathway [8]. Such transcriptional repression is an important mechanism for inhibiting the specific gene expression that is activated when β-catenin is stabilized [9]. Therefore, CTNNBIP1 would seem to possibly function as a tumor suppressor by inhibiting Wnt/β-catenin signaling, and thus blocking the oncogenic phenotype.

Previously, the importance of CTNNBIP1 has been demonstrated in embryonic development, tissue differentiation, and carcinogenesis [9,10]. Specifically, the decreased CTNNBIP1 expression levels are present in human malignant melanomas [11]. Furthermore, in colorectal tumors, the forced expression of CTNNBIP1 in the cells results in elevated levels of β-catenin, and this has been found to strongly inhibit the proliferation of these cells by inducing G2 arrest and cell death [12]. Likewise, CTNNBIP1 in human glioblastoma cells has been shown to inhibit cell proliferation, reduce cell invasion, bring about cell cycle progression arrest, and cause the induction of cell apoptosis [13]. However, little is known about the relationship between CTNNBIP1 and lung cancer. 

We have noticed that among the patients with β-catenin accumulation, there was an association with a better patient survival outcome [14,15], which in turn suggests that the antagonists of β-catenin transactivation may exert a protective role in terms of patient outcome. These results prompted us to hypothesize that CTNNBIP1 may play an important role in the repression of the β-catenin-mediated oncogenic effects present during lung cancer. To address this issue, we performed various molecular analyses using lung cancer models that targeted CTNNBIP1, in order to investigate its effect on β-catenin signaling.

## 2. Results

### 2.1. Analysis of the Factors Affecting CTNNBIP1 Gene Expression in Lung Cancer Patients

The aberrant methylation of tumor suppressor genes has been shown to be present during lung carcinogenesis [16]. To validate whether the epigenetic silencing of the CTNNBIP1 gene is involved in lung cancer, we first analyzed a cohort of patients with lung adenocarcinoma tumors, obtained from the GSE66836 and GSE66863 projects, and stored in the Gene Expression Omnibus (GEO) database [17]. This cohort involved the whole-genome DNA methylation profiling of 164 lung adenocarcinoma samples and 19 samples of matched normal lung tissue samples using the Illumina Infinium 450K array (GSE66836). The methylation levels were correlated with the gene expression levels by analysis, using the Agilent 60K mRNA expression array (GSE66863). The results showed that there was significant association between a low level of mRNA expression and the presence of DNA methylation at the CTNNBIP1 gene (Appendix A). Next, we examined whether there are any alterations in DNA methylation that affect the CTNNBIP1 gene, using samples from lung cancer patients in Taiwan. We performed a semi-quantitative reverse-transcriptase polymerase chain reaction (RT-PCR) and a quantitative RT-PCR (RT-qPCR) analysis of the mRNA extracted from 22 lung cancer patients. The results indicated that 45% (10/22) of the tumors showed a reduced level of, or the absence of, the expression of CTNNBIP1 (Figure 1A). On average, the tumor samples demonstrated lower levels of CTNNBIP1 expression than that of the normal samples (*p* < 0.001, Figure 1B). To determine whether the epigenetic alterations were involved in the gene expression of CTNNBIP1 among Taiwanese patients, we carried out DNA methylation assays targeting the CTNNBIP1 gene, using the same cohort. The results indicated that 45% (10/22) showed CTNNBIP1 promoter hypermethylation (Figure 1B). We then analyzed the correlation between the mRNA expression and promoter methylation. The low mRNA expression was significantly associated with promoter hypermethylation (*p* = 0.035; Figure 1C). Our findings support the hypothesis that promoter hypermethylation is involved in CTNNBIP1 inactivation among lung cancer patients in Taiwan.

### 2.2. CTNNBIP1 is Reactivated by 5-aza-dC in Lung Cancer Cells

In order to identify the best cell models for further investigation, we performed Western blotting to detect the protein expression of CTNNBIP1 in four human lung cancer cell lines (A549, CL1-0, CL1-5, and H1299) and in one normal cell line (MRC5). The expression of the CTNNBIP1 protein varied significantly across these cell lines (Figure 2A, left panel). The CTNNBIP1 protein was expressed at a significant level in the MRC5 and H1299 cells. However, the level of CTNNBIP1 protein was lower in the lung cancer cell lines A549, CL1-0, and CL1-5 compared with the MRC5 cell line. A quantitative RT-PCR analysis was also carried out, and this showed a significant decrease or an absence of CTNNBIP1 transcripts in the A549, CL1-0, and CL1-5 cell lines (Figure 2A, right panel).

To determine whether the promoter methylation of CTNNBIP1 is the predominant mechanism causing the loss of CTNNBIP1 expression to occur, we carried out a DNA methylation analysis of the CTNNBIP1 gene in four human lung cancer cell lines (A549, CL1-0, CL1-5, and H1299). All but one (H1299) of these lung cancer cell lines showed promoter hypermethylation, which was in sharp contrast to the situation with the normal lung cell line, MRC5, with in which the CTNNBIP1 promoter was unmethylation (Figure 2B). Next, the A549, CL1-0, and CL1-5 cells were treated with the DNA demethylating agent 5-aza-dC. Our results showed that treatment with 5-aza-dC successfully demethylated the promoter region of CTNNBIP1 gene (Figure 2C), and in the process restored the mRNA and protein expression of CTNNBIP1 in all of the lung cancer cell lines (Figure 2D,E). Taken together, the clinical correlation findings and the results from the various cell models suggest that the CTNNBIP1 expression is downregulated via DNA hypermethylation.

### 2.3. Ectopical Expression and Knockdown of CTNNBIP1 Influences the Transactivation of β-catenin in Lung Cancer Cell Lines

To verify the role of CTNNBIP1 in the transcriptional regulation of β-catenin/TCF in lung cancer, we examined the effect of ectopically-expressed CTNNBIP1s that have low levels of expression of CTNNBIP1, namely A549, CL1-0, and CL1-5 cell lines. The expression of the CTNNBIP1 protein was increased in the cells that were ectopically expressing CTNNBIP1 when these were compared with the vector control cells (Figure 3A). We further investigated whether the CTNNBIP1 expression had an effect on the transcriptional activity of β-catenin/TCF using a TCF reporter/LEF reporter assay (TOPFLASH). The TOPFLASH reporter with a mutated LEF/TCF site was used as a negative control. The β-catenin activity was significantly inhibited by the ectopic expression of CTNNBIP1 (Figure 3B), whereas there was no change in the cells containing the negative control FOPFLASH reporter. Furthermore, the expression of the β-catenin-targeted genes, MMP7, cyclin D1, and c-MYC, were found to be decreased in the cells that were ectopically expressing CTNNBIP1 (Figure 3C).

To further confirm the reciprocal relationship between the CTNNBIP1 expression and β-catenin activity in lung cancer, we examined whether the CTNNBIP1 knockdown is able to affect the β-catenin/TCF activity. The H1299 cells were transfected with siRNA-CTNNBIP1, and these were found to have a low protein expression of CTNNBIP1 compared with the cells transfected with the control siRNA (left panel, Figure 3D). We found that the β-catenin activity was significantly increased in the knock-down cells, and that the expression levels of MMP7, cyclin D1, and c-MYC were all increased in these CTNNBIP1 knockdown H1299 cells (middle and right panels, Figure 3D). Our findings suggest that the CTNNBIP1 expression plays an important role in suppressing the β-catenin transactivation in lung cancer.

### 2.4. Ectopical Expression and Knockdown of CTNNBIP1 Influences the Migration of Lung Cancer Cells

MMP7 is one of the most important β-catenin-targeted genes, and plays a crucial role in tumor cell migration and invasion [18]. Based on this, we hypothesized that CTNNBIP1 is likely to act as a migration suppressor in lung cancer. To examine whether CTNNBIP1 plays a regulatory role in cell motility, we carried out cell adhesion assays, wound healing assays, and transwell-migration assays, using various lung cancer cells that were ectopically expressing CTNNBIP1. Our findings showed that the adhesion was significantly increased 48 h after the transfection in the A549, CL1-0, and CL1-5 cells that had the ectopic expression of CTNNBIP1 (*p* = 0.042, *p* = 0.047, and *p* = 0.031, respectively; Figure 4A). Furthermore, the ectopic expression of CTNNBIP1 was found to result in a significant decrease in the migration capacity of transfected cells compared with the empty vector cells, as determined by the wound healing and transwell migration assays (Figure 4B and Figure 5A). 

To further confirm the inverse correlation between the CTNNBIP1 expression and cell motility in lung cancer, we used siRNA to generate a knockdown of CTNNBIP1 in the H1299 lung cancer cell line. The adhesion was significantly reduced in the CTNNBIP1 knockdown H1299 cells, as compared to the vector control cells (*p* = 0.036, left panel, Figure 4C). In addition, the CTNNBIP1 knockdown cells showed a significantly increased migratory capacity compared with the si-control cells, by both the wound healing assay and the transwell migration assay (right panel, Figure 4C and Figure 5B). Taken together, these results indicate that the CTNNBIP1 expression is able to affect the cell migration capacity of the various lung cancer cell model systems.

### 2.5. Correlation between the Expression of CTNNBIP1 and the Clinical Characteristics of Lung Cancer Patients

To examine whether there were associations between low levels of expression of CTNNBIP1 and the clinical characteristics of lung cancer patients, we analyzed the expression levels of CTNNBIP1 mRNA in a publicly available microarray dataset from the GSE31210 project, namely the PrognoScan database [19]. The data showed that a low level of CTNNBIP1 was correlated with the patient being a smoker (*p* = 0.008; Table 1). Importantly, the low expression of CTNNBIP1 was also significantly associated with an upward trend in the pathological stage of the lung cancer patients (*p* = 0.006; Table 1). Next, we evaluated whether the CTNNBIP1 expression was related to changes in the expression of various β-catenin-targeted genes, such as MMP7, cyclin D1, and c-MYC, among these lung cancer patients. The correlation analysis indicated that patients with an elevated level of expression of MMP7 frequently had a low expression of CTNNBIP1 (*p* = 0.001; Table 1).

To define the prognostic effects of low CTNNBIP1 in lung cancer patients, we performed a prognosis analysis using this GSE31210 project. The results showed that lower levels of CTNNBIP1 were associated with an overall poorer prognosis and a reduced relapse-free survival when 204 lung cancer patients from the GSE31210 project were analyzed (*p* = 0.011 and *p* = 0.002, respectively; Appendix A). To test whether CTNNBIP1 exerts a protective role in survival outcome, we carried out a univariate (CTNNBIP1 expression, sex, smoking habit, and tumor stage status) and a multivariate Cox regression analysis on this cohort. The results show that the low CTNNBIP1 expression was associated with a poor prognosis among these patients (*p* = 0.002; hazard ratio (HR), 2.40; 95% confidence interval (CI), 1.36–4.24 for CTNNBIP1 expression), and this remained true even after adjusting for sex, smoking habit, and tumor stage status (*p* = 0.043; HR, 1.85; 95% CI, 1.02–3.37) (Table 2). Together, these findings suggest that a low expression of CTNNBIP1 is a clinically relevant regulator of lung cancer that eventually leads to a poor prognosis.

## 3. Discussion

In an effort to better understand how alterations in the CTNNBIP1 status affect nuclear β-catenin accumulation in lung cancer, we carried out a comprehensive molecular analysis of the CTNNBIP1 gene and its relationship with the prognostic data, as well as with β-catenin activity. In the present study, we provide the first compelling evidence that CTNNBIP1 is a suppressor of lung cancer progression. The CTNNBIP1 protein is important, in that it is able to control lung cancer cell migration via the coordinated regulation of the β-catenin pathway. A low expression of CTNNBIP1 is correlated with a high level of expression of MMP7, and there is also an upward trend in terms of the pathological stage and poorer patient survival, which suggests that CTNNBIP1 may be able to serve as a prognostic biomarker for lung cancer.

Our clinical data suggested that the promoter hypermethylation is involved in the deregulation of the CTNNBIP1 gene (Figure 1 and Appendix A). Furthermore, using various lung cancer cell lines, we confirmed that the epigenetic silencing of CTNNBIP1 is linked to hypermethylation by 5-aza-dC treatment (Figure 2). This is the first study to show that promoter hypermethylation contributes to a decrease in mRNA expression of the CTNNBIP1 gene in lung cancer. Recently, evidence has suggested that such DNA methylation is generated in a programmed manner during normal aging, probably creating a predisposition towards tumorigenesis [20,21]. A previous study found that similar global hypomethylation and CpGI hypermethylation can be observed during replicative senescence (RS) and when cancers occur. A key hypothesis is that DNA methylation patterns in RS may promote tumorigenesis once cells escape RS [22]. In this earlier study, detailed genome-wide gene expression and DNA methylation patterns in both transformed and RS cells were investigated. It was found that the individual genomic regions involved in the two processes are strikingly different [23]. Specifically, the promoter CpGI hypermethylation-mediated silencing of the developmental and differentiation genes during transformation may facilitate cancer cell self-renewal and survival. In contrast, promoter hypermethylation in a senescent state mainly targets biosynthetic and metabolic genes [21,23]. Tumor suppressor genes are known to be inactivated by both genetic and epigenetic alterations [24,25]. Therefore, a detailed mutational analysis of the CTNNBIP1 gene may possibly provide a better understanding of lung tumorigenesis.

Previous studies have indicated that CTNNBIP1 may be a tumor suppressor gene, but its anti-migration properties have not yet been reported for human lung cancers. Our findings using various cellular models show, for the first time, that a loss of CTNNBIP1 brings about an increase in the cell motility of lung cancer cells (Figure 4 and Figure 5). Consistent with our findings, a low expression of CTNNBIP1 has been found in human melanomas, breast cancer, and glioblastoma [11,13,26]. In human glioblastoma, CTNNBIP1 inhibits cell proliferation and invasion, and it induces cell cycle progression arrest and cell apoptosis [13]. Furthermore, the ectopic expression of CTNNBIP1 has been found to promote the colonization of melanoma cells into the lungs of nude mice [27]. We analyzed the data from the GSE31210 project, and this showed that a low level of CTNNBIP1 is correlated with stage progression and poorer survival among lung cancer patients (Table 1 and Appendix A). These results reinforce our findings obtained using the cell model systems, and support the hypothesis that CTNNBIP1 is able to suppress the progression of lung cancer cells.

We also found that a low expression of CTNNBIP1 is significantly associated with poorer survival among Asian patients within the GSE31210 project. To understand whether the CTNNBIP1 expression plays a similar role in both Asians and Caucasians, we also performed a prognosis analysis using CTNNBIP1 on the jacob-00182-MSK project dataset for Caucasian patients [28]. As shown in Appendix A, these results also support the hypothesis that a low level of CTNNBIP1 expression is associated with poorer survival for both Asian and Caucasian patients. Notably, the Cox regression analysis reveals that a low level of CTNNBIP is the major determinant of the prognosis among lung cancer patients (Table 2). In addition, the lower level of CTNNBIP1 protein was found in poorly differentiated cases relative to well-differentiated cases, when 90 lung cancer stage IIIa patients were analyzed [29]. These findings further support the hypothesis that CTNNBIP1 is likely to be a useful prognostic factor for lung cancer.

In the present study, we provided new evidence that the CTNNBIP1 expression is able to down-regulate the β-catenin transactivation, and that this inhibits the mobility of lung cancer cells. The activation of the Wnt/β-catenin signaling influences the proliferation of cancer cells by affecting the expression of cMYC and cyclin D1 [30]. Nevertheless, we were unable to detect any relationship between the CTNNBIP1 expression, and cMYC or cyclin D1 in the clinical data. The role of CTNNBIP1 in lung cancer proliferation should be investigated more thoroughly in later studies. Recently, treatment with a new steroidal drug, NSC67657, was found to increase the expression of CTNNBIP1, decrease the expression of β-catenin, and induce the monocytic differentiation of HL-60 cells [31]. In addition, epigenetic control therapy may also have a place as a potential adjuvant treatment for lung cancer [32]. Therefore, strategies that increase the CTNNBIP1 expression may be useful as methods of cancer prevention and treatment. The search for antagonists or agonists of CTNNBIP1 may also lead to the discovery of new compounds that may potentially be useful when treating lung cancer.

## 4. Materials and Methods

### 4.1. Subjects

Paired tumor and normal lung tissues were obtained from 22 lung cancer patients prospectively recruited at the Chia-Yi Christian Hospital between 2014 and 2017, after appropriate institutional review board permission had been obtained. Informed consent was obtained from all of the patients (CYCH IRB no: 102072). The inclusion criteria for the patient sample collection were that the patient was diagnosed with lung cancer, and that the treatment required surgical resection of the tumor. The exclusion criterion for the patient sample collection were that, following surgery, any patients who were diagnosed by pathology as having a benign tumor of the lung were excluded from the present study. For the methylation assays, the genomic DNA from primary tumor tissue samples was extracted using proteinase K digestion, which was followed by the phenol–chloroform extraction. For the RNA expression assays, the total RNA was extracted from the paired tumor and normal tissue samples using Trizol reagent (Invitrogen, Carlsbad, CA, USA). The cDNA was then synthesized using SuperScriptTM reverse transcriptase (Invitrogen), according to the manufacturer’s instructions.

### 4.2. Cell Culture

The lung cancer cell line A549 was purchased from the Bioresource Collection and Research Center. The isogenic human lung cancer cell lines, CL1-0 and CL1-5, with a low and high motility, respectively, were kindly provided by Dr. Pan-Chyr Yang at the Department of Internal Medicine, National Taiwan University, Taiwan [33]. The normal human lung cell line, MRC5, and human lung cancer cell lines, H1299, were kindly provided by Dr. Yi-Ching Wang at the Institute of Pharmacology, National Cheng Kung University-Tainan. All of the cell lines were maintained in Dulbecco’s Modified Eagle Medium (DMEM; pH 7.4, Invitrogen) with 10% Fetal Bovine Serum (FBS; Invitrogen) and 1% penicillin/streptomycin (100 units/mL penicillin and 100 μg/mL streptomycin, Invitrogen). All of the cell lines were incubated at 37 °C, in a humidified atmosphere containing 5% CO_2_/95% air.

### 4.3. RT-PCR Analysis

The CTNNBIP1 mRNA expression levels were measured by a multiplex RT-PCR analysis using the GADPH gene as an internal control. The primers used are listed in Appendix A. The reactions were carried out in a final volume of 25 µL with 1 µL of cDNA and 0.25 *p*mol of primers, using a Bio-Rad MyCycler Thermal Cycler (Bio-Rad, CA, USA). The relative levels of the gene expression were calculated as previously described [34].

### 4.4. RT-qPCR Assays

Using the various cell models, RT-qPCR targeting the CTNNBIP1, MMP7, cyclin D1, and c-MYC mRNA expression was performed using the β-actin gene as the internal control. The primers used are listed in Appendix A. The mRNA level of CTNNBIP1, MMP7, cyclin D1, and c-MYC were calculated using 2^−∆∆*C*t^ (^Δ^Ct = C_t-gene_ − C_t-β-actin_).

### 4.5. Western Blot Analysis

The cells were lysed, and the resulting lysates were centrifuged. Next, an SDS gel loading buffer (60 mM Tris base, 2% SDS, 10% glycerol, and 5% β-mercaptoethanol) was added. Samples containing 50 µg of protein were then separated by 8% SDS-PAGE, followed by electro-blotting onto Immobilon-P membranes (Millipore, Bedford, MA, USA). Immunoblotting was performed using antibodies against CTNNBIP1 (1:500; R&D Systems, Minneapolis, MN, USA). β-actin (1:5000) (GeneTex, Irvine, CA, USA) was used as the loading control.

### 4.6. Methylation-Specific PCR (MSP) Assay

The methylation status within the promoter region of the CTNNBIP1 gene was determined by chemical treatment with sodium bisulfite, and the subsequent MSP analysis. Positive control samples with unmethylated lymphocyte DNA and SssI methyltransferase-treated methylated DNA (M reaction) were included in each PCR set. The primers used are listed in Appendix A. Bisulfite-modified DNA (100 ng) was amplified by PCR (35 cycles for U reaction; 35 cycles for M reaction) using an annealing temperature of 55 °C for both the U and M reactions. The hypermethylated genes were defined as those that produced a larger amount of amplicon from the M reaction than from the U reaction for a given sample.

### 4.7. 5-aza-2′-Deoxycytidine (5-aza-dC) Treatment

Cancer cells (1 × 10^5^ per dish) were plated in a 100-mm culture dish the day before treatment. Next, the cells were treated with 20 μmol/L 5-aza-dC (Sigma, St Louis, MO, USA) for three doubling times, and then the cells were harvested for MSP, RT-qPCR, and Western blot analysis.

### 4.8. Knockdown and Ectopic Expression

The siRNA-CTNNBIP1/control-siRNA and pCMV6/pCMV6-CTNNBIP1 vectors were obtained from OriGene (OriGene, Rockville, MD, USA). The cancer cells (1 × 10^5^) were transfected with 5 μg of siRNA-CTNNBIP1 or pCMV6-CTNNBIP1 using ExGen 500 transfection reagent (Fermentas, Hanover, MD, USA), as recommended by the manufacturer. After incubation, the cells were subjected to RT-qPCR and Western blot analysis.

### 4.9. Luciferase Reporter Assay

The TCF luciferase constructs, containing the wild-type (pTOPFLASH) or the mutant (pFOPFLASH) TCF binding sites (Upstate, Lake Placid, NY, USA), were co-transfected with an internal control (pRLTK Renilla luciferase vector; Promega, Madison, WI, USA) into cells containing either pCMV6-CTNNBIP1 or si-CTNNBIP1 (5 × 10^5^ cells). The Firefly (TOPFLASH or FOPFLASH) luciferase activity was normalized against the Renilla luciferase activity. The TOPFLASH activity was also normalized against the FOPFLASH activity.

### 4.10. Wound Healing Assay

The wound healing assay was carried out using the ibidi culture insert system (Applied Biophysics, Troy, NY, USA), with A549, CL1-0, CL1-5, and H1299 cells that had been transfected with either an expression plasmid or an appropriate si-RNA oligo targeting CTNNBIP1. The cells were starved for 16 h before conducting the migration experiment. A cell-free gap of 500 μm was created by removing the insert. The cell-free gap area that remained after culturing for 16 h was determined using ImageJ software, by measuring the width of the remaining open wound and assessing the rate of wound closure. Three independent experiments were performed.

### 4.11. Transwell Migration Assay

Transwell assays were performed in order to determine the migration and invasion ability of the CTNNBIP1 overexpression cells and the vector control cells. A similar analysis was also performed using CTNNBIP1 knockdown cells and si-control cells. The transwell system (Falcon, Bedford, MA, USA) consisted of upper and lower chambers, separated by a layer of millipore membrane with pore size of 8 μm. The cells seeded in the upper chamber could migrate through the membrane to the lower chamber. About 5 × 10^5^ cells were seeded into the upper chamber of the transwell with a serum-free DMEM medium, and the lower chamber containing a 10% FBS/DMEM attractant medium. After incubation for 24 h, the cells attached to the reverse phase of the membrane were stained by crystal violet, or were observed for cells with fluorescence, and were counted under a microscope in six randomly selected fields. The experiment was carried out three times in order to reduce the possible effects of biological variability.

### 4.12. Adhesion Assay

An appropriate number of six-well microplates were pre-coated with various extracellular matrix proteins, namely collagen type I, 4 μg/well (Sigma). The wells were then blocked using 1% bovine serum albumin at 4 °C. The cells (5 × 10^4^ cells/well) of the knockdown cells, overexpression cells, or control cells were then seeded onto the pre-coated wells, and then they were allowed to recover for 2 h in a medium containing 20% FBS at 37° C, in an incubator with rotation. Next, the seeded cells were incubated for 30 min at 37 °C with 5% CO_2_. Any unattached cells were then removed by PBS washing three times. The remaining attached cells were fixed and stained using 1% crystal violet/MeOH for 10 min at room temperature. Finally, they were lysed using DMSO. The absorbance at 590 nm is known to be correlated with the number of cells attached to the coated ECM wells, and this was measured. The experiment was carried out three times in order to reduce the possible effects of biological variability.

### 4.13. Statistical Analysis

Pearson’s χ^2^ test was used to compare the frequency of the protein alteration among DNA methylation and the mRNA expression, and the clinical parameters of the lung cancer patients at different disease stages. The RT-qPCR, β-catenin activity, wound healing, and transwell assay were analyzed by Student’s t-test or two-way analysis of variance (ANOVA) analysis. The overall survival curves were calculated using the Kaplan–Meier method, and then a comparison was performed using the log-rank test. The univariate and multivariate analyses were estimated using Cox proportional hazards regression. A value of *p* < 0.05 was considered statistically significant. SPSS version 19.0 (IBM-SPSS, Chicago, IL, USA) was used for all of the statistical analyses.

## Figures and Tables

**Figure 1 ijms-20-05684-f001:**
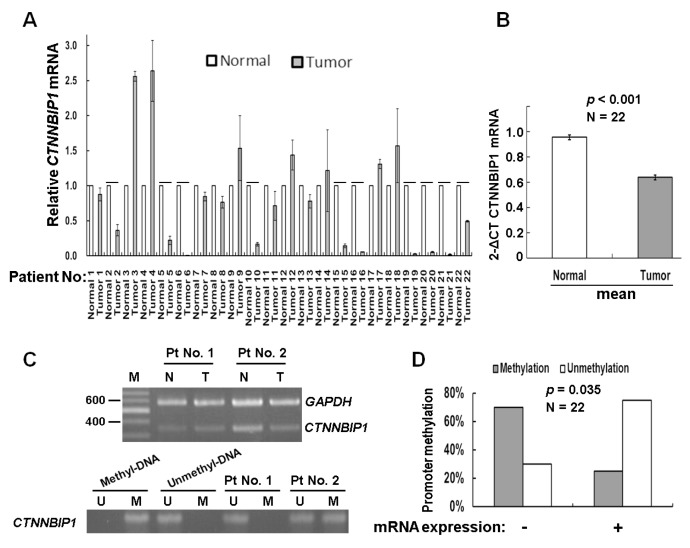
Changes in β-catenin-interacting protein 1 (CTNNBIP1) gene expression and DNA methylation among lung cancer patients. (**A**) The mRNA expression of the CTNNBIP1 gene in 22 lung cancer patients by quantitative RT-PCR analysis. Data are presented as the mean ± standard deviation (SD; *n* = 3). Tumor expression levels < 50% that of the normal cells were deemed to have a low expression. “-” indicates a low expression of CTNNBIP1. (**B**) On average, tumor samples showed a lower CTNNBIP1 expression than the paired normal tissue (*p* < 0.001, by two-way analysis of variance (ANOVA) test). Data are presented as the mean ± SD (*n* = 3). (**C**) Semi-quantitative RT-PCR (upper panel) and MSP (lower panel) were conducted to analyze the mRNA expression levels of CTNNBIP1 and the promoter methylation at CTNNBIP1. N—control samples; T—tumor tissue samples. The primer sets used for amplification are designated as “U” for the unmethylated genes, or “M” for the methylated genes. (**D**) A negative correlation between the RNA expression and CTNNBIP1 DNA methylation was found for the 22 lung cancer patients (*p* = 0.035, by the Pearson’s χ^2^ test). “+” indicates the mRNA expression, while “-“ represents a low expression. The concordant group is the “RNA-/unmethylation” group, and the discordant group is the “RNA+/methylation” group.

**Figure 2 ijms-20-05684-f002:**
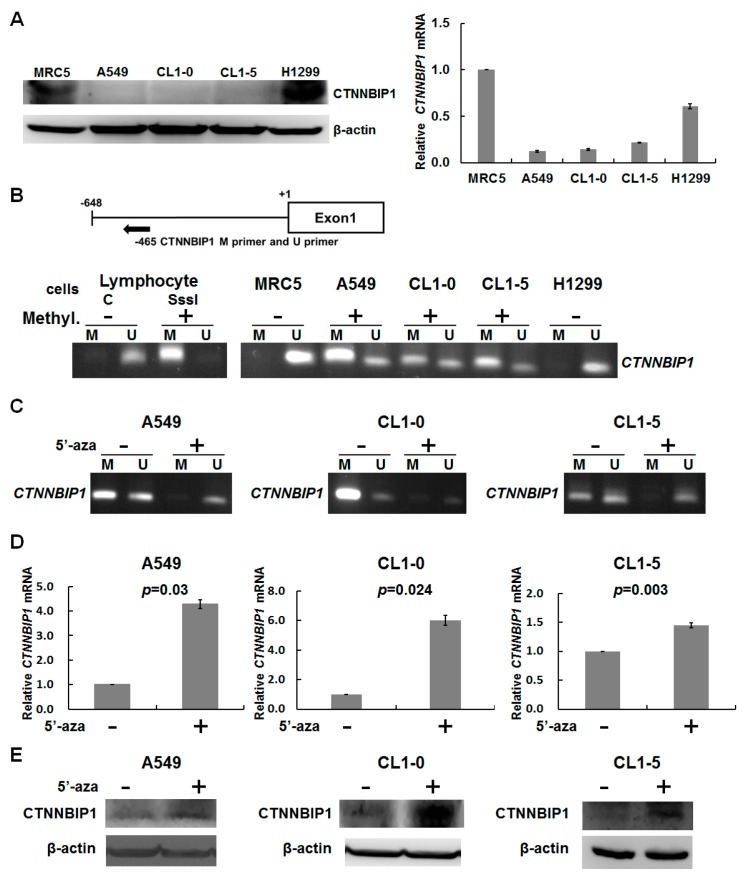
The promoter methylation of the CTNNBIP1 gene in lung cancer cells. (**A**) The distribution of the CTNNBIP1 protein and mRNA across normal and lung cancer cell lines. (**B**) A schematic representation of the genomic structure of the CTNNBIP1 locus shows the positions of the primers used for the MSP assay (upper panel). MSP analysis of the CTNNBIP1 gene in the normal lung cell line MRC5 and in various lung cancer cell lines, namely, A549, CL1-0, CL1-5, and H1299 (lower panel). (**C**) MSP analysis of the CTNNBIP1 gene in the lung cancer cell lines A549, CL1-0, and CL1-5 after 5′-aza-dC treatment. Positive control samples with unmethylated lymphocyte DNA (U reaction) and SssI methyltransferase-treated methylated DNA (M reaction) were included in the MSP assay. (**D**) Quantitative RT-PCR (*p* = 0.03 in A549, *p* = 0.024 in CL1-0, and *p* = 0.03 in CL1-5, by a two-tailed paired t-test.) and (**E**) Western blot analysis of CTNNBIP1 mRNA expression and protein expression in the 5′-aza-dC-treated A549, CL1-0, and CL1-5 cell lines. “+” indicates 5′-aza treatment, while “−“ no treatment. “M” for the methylated genesor, “U” for the unmethylated genes. Quantitative data are presented as the mean ± SD from three independent experiments.

**Figure 3 ijms-20-05684-f003:**
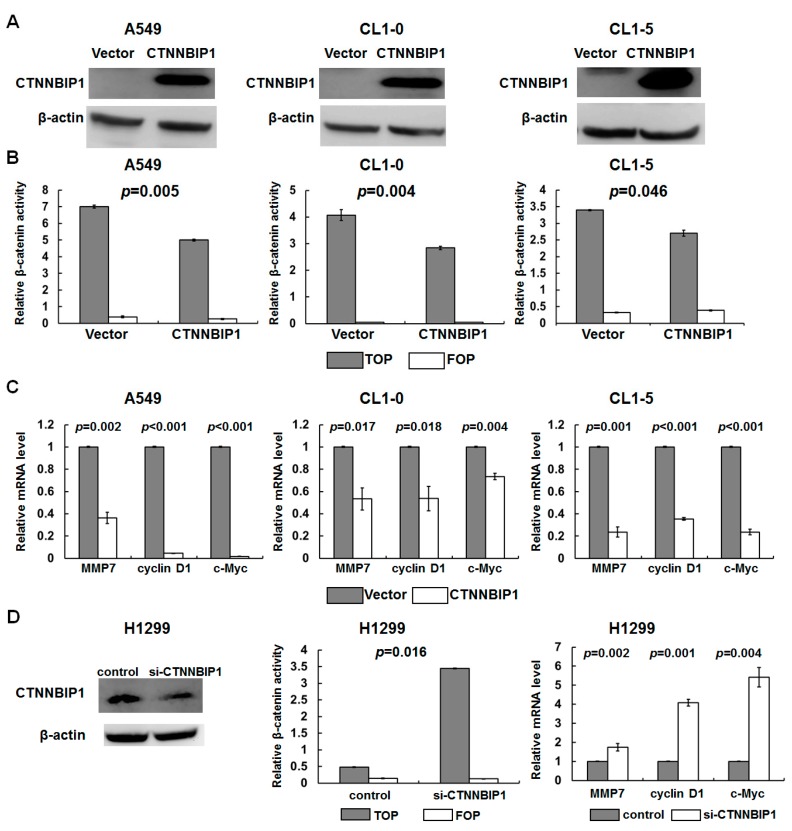
The effect of the ectopic expression and knockdown of CTNNBIP1 on β-catenin transactivation in lung cancer cells. (**A**) Western blot analysis showed an increase in the CTNNBIP1 expression when there was ectopic expression of CTNNBIP1 in the A549, CL1-0, and CL1-5 cells. (**B**) The cells expressing an empty vector or CTNNBIP1 were transfected with the TOPFLASH reporter. Decreased β-catenin transactivation in A549, CL1-0, and CL1-5 cell lines was present in cells ectopically expressing CTNNBIP1. (**C**) The quantitative RT-PCR analysis of the MMP7, cyclin D1 and c-Myc mRNA levels in the cells ectopically expressing CTNNBIP1. (**D**) Inverse correlation of the CTNNBIP1 expression levels with CTNNBIP1 knockdown using H1299 cells. The data are representative of three independent experiments. The *p*-value for each analysis is provided. The quantitative data are presented as the mean ± SD from three independent experiments. The two-tailed paired t-test was used to compare the frequency between the ectopic expression of the CTNNBIP1 and β-catenin transactivation.

**Figure 4 ijms-20-05684-f004:**
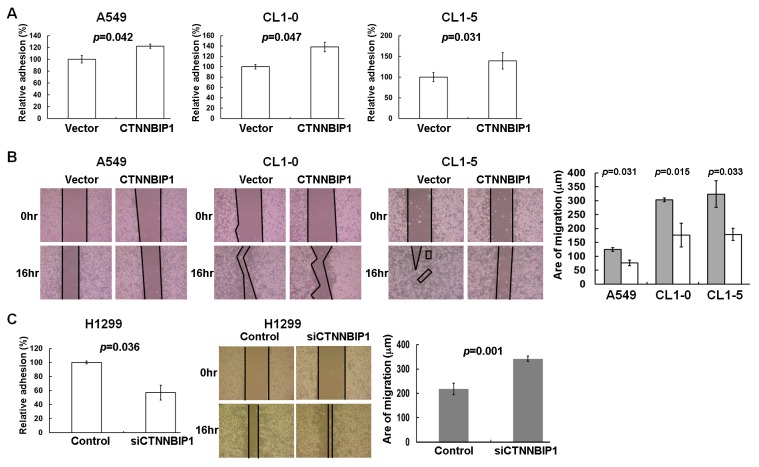
The effect of the ectopic expression and knockdown of CTNNBIP1 on the cell migration ability of lung cancer cells. (**A**) The adhesion ability was increased in A549, CL1-0, and CL1-5 cells when there was ectopic expression of CTNNBIP1 after transfection compared with the empty vector controls. (**B**) The ectopic expression of CTNNBIP1 decreases the cell motility, as assessed by the wound healing assay. The cells in the culture were monitored for their ability to migrate into the created wound gap. The created wound gap was photographed at 0 and 16 h. The wound healing images demonstrated that the A549, CL1-0, and CL1-5 cells that were ectopically expressing CTNNBIP1 migrated much slower than the vector control cells. Quantitative wound healing data for the various lung cancer cell lines is provided in the right panel. (**C**) Inverse correlation of the CTNNBIP1 expression level after CTNNBIP1 knockdown in the H1299 cells. The *p*-value for each analysis is provided by a two-tailed paired t-test. Quantitative data are presented as the mean ± SD from three independent experiments.

**Figure 5 ijms-20-05684-f005:**
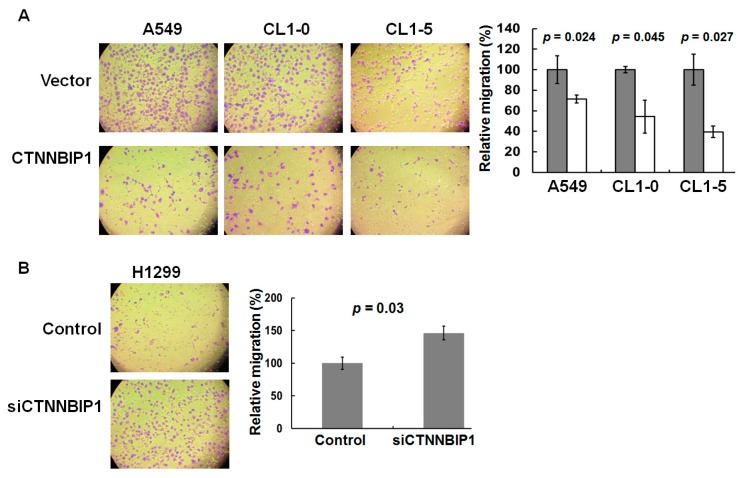
The ectopic expression and knockdown of CTNNBIP1 influences the lung cancer cell migration ability by the transwell migration assay. (**A**) Ectopically-expressed CTNNBIP1 decreases cell motility in the lung cancer cell lines of the A549, CL1-0, and CL1-5 (left panel); the quantitative data is shown in the right panel. (**B**) Inverse correlation between the CTNNBIP1 expression level and cell migration when there is a CTNNBIP1 knockdown in the H1299 cells. A *p*-value for each analysis is provided by a two-tailed paired t-test. The quantitative data are presented as the mean ± SD from three independent experiments.

**Table 1 ijms-20-05684-t001:** CTNNBIP1 expression relative to the clinicopathological parameters of lung cancer patients ^a^.

Characteristics		*CTNNBIP1* Expression			
		Total	+	− (%)	*p*-Value ^b^
Overall		204	162	42 (20.6)	
Sex	Female	109	92	17 (15.6)	0.059
	Male	95	70	25 (26.3)	
Smoker	No	105	91	14 (13.3)	**0.008**
	Yes	99	71	28 (28.3)	
Tumor stage	IA	109	95	14 (12.8)	**0.006**
	IB	53	40	13 (24.5)	
	II	42	27	15 (35.7)	
MMP7	high	40	24	16 (40.0)	**0.001**
	low	164	138	26 (15.9)	
Cyclin D1	high	34	26	8 (23.5)	0.642
	low	170	136	34 (20.0)	
c-MYC	high	61	49	12 (19.7)	0.833
	low	143	113	30 (21.0)	

^a^ These results were analyzed from the data of the lung cancer patients of the publicly available microarray data in the GSE31210 project from the PrognoScan database. ^b^ Bold values indicate statistical significance (*p* < 0.05). “+” indicates the mRNA expression, while “−“ represents a low expression.

**Table 2 ijms-20-05684-t002:** Univariate and multivariate analysis by a Cox regression model ^a^.

Characteristics	Univariate Analysis		Multivariate Analysis	
	HR (95%CI)	*p* Value *	HR (95%CI)	*p* Value ^b^
CTNNBIP1				
Expression	1.00		1.00	
Low	2.40 (1.36–4.24)	**0.002**	1.85 (1.02–3.37)	**0.043**
Sex				
Female	1.00		1.00	
Male	1.40 (0.82–2.38)	0.220	1.18 (0.57–2.41)	0.659
Smokier				
No	1.00		1.00	
Yes	1.43 (0.84–2.44)	0.190	1.07 (0.52–2.22)	0.853
Tumor stage				
I	1.00		1.00	
II	3.44 (1.98–6.00)	**<0.001**	2.91 (1.63–5.18)	**<0.001**

^a^ These results were analyzed from data of the lung cancer patients of the publicly available microarray data in GSE31210 project by the PrognoScan database. ^b^ Bold values indicate statistical significance (*p* < 0.05). CI—confidence interval; HR—hazard ratio.

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
