# Peer review of "The Alteration of CTNNBIP1 in Lung Cancer"

_ijms, 2019, doi:10.3390/ijms20225684_

Round 1
Reviewer 1 Report
The work evaluating the alteration of CTNNBIP1 in lung cancer, specifically addressing the methylation burden in its promoter, is well written. However, the results are not clearly presented, thus difficult their interpretation. More details should be provided in the methodology and in the result sections.
It would facilitate the understanding of the analysis if the authors provide more background information about the GEO database. What do “methylation profiling” and “expression profiling” mean? What was the criterion applied to differentiate these 2 groups? Data from S1 is available, but here it was analysed for the first time in terms of association of methylation and CTNNBIP1 expression? Please clarify.
Please provide the Ethics approval number that permits the usage of the normal and cancer samples, from the 22 lung cancer patients, in this study.
Indicate the selection criteria that allowed the inclusion or exclusion of patients in the study.
Please indicate at what the passage number the cells were used. Also discuss how the methylation status could be affected by the cell line passage, as epigenetic changes are influenced by aging.
Please provide the full WB gel of Figure S1, including the molecular weight markers (ladder).
In Figure 1, it is not clear what “1” and “2” mean. Bar graph should show standard deviation and a statistical analysis, 2-way ANOVA, should be performed.
In the legend of each figure,
- Please indicate the statistical analysis performed.
- Please indicate if the error bars are expressed as SEM or SD.
- Please indicate if the assays were performed in duplicate, triplicate, etc. How many biological replicate (independent cultures) were assessed, N3?
Author Response
First, we would like to thank the editor and the reviewers for a very thorough and critical examination of our manuscript. We also appreciate the helpful suggestions and comments. They have been incorporated into our revised manuscript. Responses to specific comments are listed below.
Response to comments by reviewer 1:
The work evaluating the alteration of CTNNBIP1 in lung cancer, specifically addressing the methylation burden in its promoter, is well written.
It would facilitate the understanding of the analysis if the authors provide more background information about the GEO database. What do “methylation profiling” and “expression profiling” mean? What was the criterion applied to differentiate these 2 groups? Data from S1 is available, but here it was analysed for the first time in terms of association of methylation and CTNNBIP1 expression? Please clarify.
In order to perform our results more clearly, we have described the detailed background information about the GEO database in the Results section and the revised Table S1. In our study, the expression value of CTNNBIP1 mRNA below the mean value was indicated low RNA expression. Since loss of TSG function occurs via the inactivation of two alleles. Therefore, we increased the criteria of the methylation value of CTNNBIP1 above mean+1SD was indicated DNA methylation.
Please provide the Ethics approval number that permits the usage of the normal and cancer samples, from the 22 lung cancer patients, in this study.
The Ethics approval number CYCH-IRB 102072 has been indicated in Materials and Methods section.
Indicate the selection criteria that allowed the inclusion or exclusion of patients in the study.
Thanks for the suggestion. Inclusion criteria for the patient sample collection were as follows: The patient was diagnosed with lung cancer and the treatment required surgical resection of the tumor. Exclusion criteria for the patient sample collection were as follows: Following surgery, patients who were diagnosed by pathology as benign tumor in the lung were excluded from the present study. The inclusion or exclusion of patients in this study has been indicated in Materials and Methods section.
Please indicate at what the passage number the cells were used. Also discuss how the methylation status could be affected by the cell line passage, as epigenetic changes are influenced by aging.
We thank the reviewer for pointing out the importance of the methylation status and aging. DNA methylation is important, as it is known to be abnormal in all forms of cancer. A key hypothesis is that DNA methylation patterns in Replicative Senescence (RS) may promote tumorigenesis once cells escaping RS. Recently, Xie et al. (2018) found that the individual genomic regions involved in the RS and tumorigenesis processes are strikingly different. We have included a detailed discussion of the methylation and aging in the revised Discussion section. The A549 cell line (P=95) was purchased from the Bioresource Collection and Research Center. The passage number 95 of A549 cells was used in this study. The CL1-0 and CL1-5 cells were kindly provided by Dr. Pan-Chyr Yang.
Please provide the full WB gel of Figure S1, including the molecular weight markers (ladder).
The full WB gel of Figure S1were as follows:
In Figure 1, it is not clear what “1” and “2” mean. Bar graph should show standard deviation and a statistical analysis, 2-way ANOVA, should be performed.
Thanks for the suggestion. The “1” and “2” mean the patient number. Indeed the 2-way ANOVA is clearer for relation analysis. The normal and tumor tissue of 22 patients and the standard deviation are analyzed by the 2-way ANOVA test has been indicated in the revised Figure 1B.
In the legend of each figure,
- Please indicate the statistical analysis performed.
- Please indicate if the error bars are expressed as SEM or SD.
- Please indicate if the assays were performed in duplicate, triplicate, etc. How many biological replicate (independent cultures) were assessed, N3?
Thanks for the suggestion. We have indicated the statistical analysis, SD, and the number of biological replicate in each revised figure legend.

Reviewer 2 Report
The authors have investigated the role of CTNNBIP1 in lung cancer. CTNNBIP1 prevents expression of B-catenin target genes by acting as a competitor of B-catenin with TCF in the nucleus. CTNNBIP1 has been identified as a tumor suppressor in other cancer types, so the rationale is strong to look in lung cancer as well. Overall, this manuscript covers an important issue, is well written, and the science is well designed. There are a few places for improvement, listed below:
Major concerns:
In Figure 1, all 22 samples should be shown, and the graphs should be average with STDEV of all samples (normal and tumor). In addition, the correlation between RT-PCR and methylation needs to be shown in a correlation plot. In Figure 2B, it's unclear why there is no increase in the unmethylated genes after treatment with 5-aza. It's possible that the concentration used was high enough to kill the cells, so some clarification should be provided. The blots in Figure 3A (vector transfected) should be the same as in Figure 2D (without 5-aza). There is a concern that in 3A, there is no expression with the vector, but in 2D, there is expression in both the CL1-0 and CL1-5 cells before treatment with 5-aza. There is likely a change in cell proliferation (if you are altering B-catenin mediated transcription). It's not mentioned if this was controlled for in the migration assays (both Figures 4 and 5). If nothing was done to control for proliferation, these assays need to be repeated (either with mitomycin or low serum media).Minor concerns:
The paper needs editing for the English language. The data in Figure S1 should be a part of the manuscript. The rationale for using lymphocytes in Figure 2A is unclear. There are multiple adhesion molecules described in the methods, but there is no indication of what is being presented. Legends need to be included in all figures (ie, 4 and 5). Higher quality (magnification) images are needed for Figure 5.Author Response
First, we would like to thank the editor and the reviewers for a very thorough and critical examination of our manuscript. We also appreciate the helpful suggestions and comments. They have been incorporated into our revised manuscript. Responses to specific comments are listed below.
Response to comments by reviewer 2:
The authors have investigated the role of CTNNBIP1 in lung cancer. CTNNBIP1 prevents expression of B-catenin target genes by acting as a competitor of B-catenin with TCF in the nucleus. CTNNBIP1 has been identified as a tumor suppressor in other cancer types, so the rationale is strong to look in lung cancer as well. Overall, this manuscript covers an important issue, is well written, and the science is well designed.
Major concerns:
In Figure 1, all 22 samples should be shown, and the graphs should be average with STDEV of all samples (normal and tumor). In addition, the correlation between RT-PCR and methylation needs to be shown in a correlation plot.We thank the reviewer's suggestion. The data are shown below and are included in the revised Figure 1A. Pearson’s χ2 test is used to compare the correlation between the mRNA expression and DNA methylation of CTNNBIP1 (revised Figure 1D).
In Figure 2B, it's unclear why there is no increase in the unmethylated genes after treatment with 5-aza. It's possible that the concentration used was high enough to kill the cells, so some clarification should be provided.
The 5-aza is an inhibitor of maintenance MTase, DNMT1. After DNA replication, hemimethylated sites are remethylated by the DNMT1 restoring pre-existing patterns of methylation. Therefore, it is speculated that the state of hemimethylation may cause a decrease in M primer performance, but does not increase U primer performance.
The blots in Figure 3A (vector transfected) should be the same as in Figure 2D (without 5-aza). There is a concern that in 3A, there is no expression with the vector, but in 2D, there is expression in both the CL1-0 and CL1-5 cells before treatment with 5-aza.
For this point, our data showed the low level of CTNNBIP1 transcripts in the A549, CL1-0, and CL1-5 cell lines by quantitative RT–PCR analysis (Figure S1B). It is indicated that among the three cell lines, the CTNNBIP1 gene has a small endogenous expression. Therefore, a little protein performance is seen in Figure 2D. Since the CTNNBIP1 protein is overexpressed in Figure 3A, high signal intensity masks the data, thus weakening the performance of the control vector.
There is likely a change in cell proliferation (if you are altering B-catenin mediated transcription). It's not mentioned if this was controlled for in the migration assays (both Figures 4 and 5). If nothing was done to control for proliferation, these assays need to be repeated (either with mitomycin or low serum media).
We starve the cells for 16 hours before conducting a migration experiment. It is indicated in Materials and Methods section.
Minor concerns:
The paper needs editing for the English language. The data in Figure S1 should be a part of the manuscript. The rationale for using lymphocytes in Figure 2A is unclear. There are multiple adhesion molecules described in the methods, but there is no indication of what is being presented. Legends need to be included in all figures (ie, 4 and 5). Higher quality (magnification) images are needed for Figure 5.
According to the suggestions, we revised manuscript has been proofread by the English editor. We already put Figure S1 in the revised Figure 2. Thanks for your kindness. We have revised the multiple adhesion molecules of adhesion assay in Materials and Methods section. Lymphocytes with unmethylated CTNNBIP1 DNA were used as control samples. We have replaced the higher quality images in the revised Figure 5.
Round 2
Reviewer 1 Report
Authors have addressed the comments and included the clarifications in the manuscript.
Therefore, I recommend its publication.
Author Response
We thank the reviewer for the comment.
Reviewer 2 Report
Manuscript can be accepted after minor editing.
Author Response
Thanks for the suggestion. We have been making minor changes in this manuscript.